# Enzymatic Characterization of the Isocitrate Dehydrogenase with Dual Coenzyme Specificity from the Marine Bacterium *Umbonibacter marinipuiceus*

**DOI:** 10.3390/ijms241411428

**Published:** 2023-07-13

**Authors:** Mingjie Bian, Jiaxin Zhao, Wenqiang Xu, Xueyang Han, Xuefei Chen, Peng Wang, Guoping Zhu

**Affiliations:** Anhui Provincial Key Laboratory of Molecular Enzymology and Mechanism of Major Diseases, Key Laboratory of Biomedicine in Gene Diseases and Health of Anhui Higher Education Institutes, Anhui Normal University, Wuhu 241000, China

**Keywords:** isocitrate dehydrogenase, dual coenzyme specificity, biochemical characterization, kinetics, molecular evolution, *Umbonibacter marinipuiceus*

## Abstract

Isocitrate dehydrogenase (IDH) can be divided into NAD^+^-dependent and NADP^+^-dependent types based on the coenzyme specificity. It is worth noting that some IDHs exhibit dual coenzyme specificity characteristics. Herein, a dual coenzyme-dependent IDH from *Umbonibacter Marinipuiceus* (UmIDH) was expressed, purified, and identified in detail for the first time. SDS-PAGE and Gel filtration chromatography analyses showed that UmIDH is an 84.7 kDa homodimer in solution. The *K*_m_ values for NAD^+^ and NADP^+^ are 1800.0 ± 64.4 μM and 1167.7 ± 113.0 μM in the presence of Mn^2+^, respectively. Meanwhile, the catalytic efficiency (*k*_cat_/*K*_m_) of UmIDH is only 2.3-fold greater for NADP^+^ than NAD^+^. The maximal activity for UmIDH occurred at pH 8.5 (with Mn^2+^) or pH 8.7 (with Mg^2+^) and at 35 °C (with Mn^2+^ or Mg^2+^). Heat inactivation assay revealed that UmIDH sustained 50% of maximal activity after incubation at 57 °C for 20 min with either Mn^2+^ or Mg^2+^. Moreover, three putative core coenzyme binding residues (R345, L346, and V352) of UmIDH were evaluated by site-directed mutagenesis. This recent work identified a unique dual coenzyme-dependent IDH and achieved the groundbreaking bidirectional modification of this specific IDH’s coenzyme dependence for the first time. This provides not only a reference for the study of dual coenzyme-dependent IDH, but also a basis for the investigation of the coenzyme-specific evolutionary mechanisms of IDH.

## 1. Introduction

Isocitrate dehydrogenase (IDH) is a key limiting enzyme in the tricarboxylic acid cycle (TCA cycle), which catalyzes the oxidation and decarboxylation of isocitrate (ICT) to alpha-ketoglutarate (α-KG), NAD(P)H, and CO_2_ with the participation of the coenzymes NAD^+^/NADP^+^ and the activator Mn^2+^ or Mg^2+^ [1]. NAD^+^-dependent IDH (EC 1.1.1.41, NAD-IDH) produces NADH, which participates in energy metabolism. It is mainly distributed in the mitochondria of eukaryotic cells, as well as in a few bacteria and archaea [2]. NADP^+^-dependent IDH (EC 1.1.1.42, NADP-IDH) is widely distributed in various organisms, and can generate NADPH through catalytic action, which provides reducing power for biosynthesis [3]. Mutations in the relevant sites of cytoplasmic and mitochondrial NADP-IDH in human cells can lead to an increased occurrence rate of cancer [4,5,6].

The IDH family belongs to the β-decarboxylating dehydrogenase superfamily, can be divided into three subfamilies (Type I, Type II, and Type III) based on phylogenetic analysis [7,8,9]. Type I IDHs comprise the vast majority of bacterial and all archaeal homodimeric NAD(P)-IDHs, mitochondrial hetero-oligomeric NAD-IDHs, and bacterial homotetrameric NAD-IDHs. Type II IDHs contain bacterial homodimeric NADP-IDHs and homohexameric NAD-IDHs and eukaryotic homodimeric NAD(P)-IDHs. Type III IDHs are mainly composed of bacterial monomeric NAD(P)-IDHs and a few marine archaeon monomeric NADP-IDHs. A group of structurally unique homodimeric or homotetrameric IDHs with monomer-liked subunits are also clustered into the type III subfamily. Notably, phylogenetic analysis revealed that three IDH subfamilies evolved independently.

IDH plays a crucial role in studying molecular evolution and it has been a significant topic for research and analysis. NADP-IDH predominates in the IDH family, and research on Type I IDHs suggested that NAD^+^-dependency is an ancestral trait of IDH, while NADP^+^-dependency is an adaptation phenotype [10,11,12]. The evolution from NAD-IDH to NADP-IDH occurred approximately 3.5 billion years ago, and allowed bacteria to overcome the stress of carbon limitation [13]. Coenzyme-specific modification experiments based on site-directed mutagenesis had simulated this evolutionary mechanism, and to some extent, validated this hypothesis. Therefore, as the transition state in the evolution of coenzyme specificity of IDH, dual coenzyme-dependent IDH may have phylogenetic implications.

However, there is limited research on dual coenzyme IDH. Among the reported dual coenzyme IDHs, there are substantial variations in preference for either of the two coenzymes, with one of the coenzymes lacking physiological significance due to its extremely low catalytic efficiency [14,15,16]. In comparison, the *Methylobacillus flagellatus* IDH (MfIDH), belonging to the Type I subfamily, exhibits relatively comparable affinity for and catalytic efficiency with either coenzyme. The *K*_m_ values of MfIDH for NAD^+^ and NADP^+^ were 113 μM and 184 μM, respectively, while the specificity (*k*_cat_/*K*_m_) was only five times higher for NAD^+^ than for NADP^+^ [17]. Another pyridine nucleotide-dependent oxidase, an oxidase from *Lactobacillus reuteri* (LreNox) [18], also exhibits unique dual substrate specificity, showing equal catalytic efficiency with NADH or NADPH.

*Umbonibacterium marinipuiceus* is a gram-negative bacterium that was originally isolated from marine mollusc costellae collected in the Sea of Japan [19]. It exhibits an aerobic lifestyle, appears as rod-shaped cells, lacks motility, and displays sensitivity to slight changes in salinity. Moreover, it demonstrates resilience to cold temperatures but is incapable of degrading most of the complex polysaccharides and polar lipids that were tested. The growth of *U. marinipuiceus* necessitates the presence of sodium chloride (NaCl). It thrives within a range of 2% to 5% NaCl, with the most favorable concentration being approximately 2.5% to 3%. In this study, we conducted overexpression, purification, and detailed biochemical characterization of a type I homodimeric IDH from *U. marinipuiceus* (UmIDH), which will provide reference for future research on dual coenzyme-dependent IDHs.

## 2. Results

### 2.1. Bioinformatics Analysis

The UmIDH gene has a total length of 1272 base pairs (bp) and encodes a protein consisting of 420 amino acids, which conforms to the characteristics of Type I IDH. The sequence identities of UmIDH with typical Type I subfamily members NADP^+^-IDH from *E. coli* (EcIDH), NAD^+^-IDH from *A. thiooxidans* (AtIDH), and dual coenzyme-dependent MfIDH are 63.6%, 54.6%, and 61.0%, respectively. In this study, we performed a detailed phylogenetic analysis to reconstruct the evolutionary tree of the IDH family (Figure 1). It is evident that UmIDH belongs to the Type I subfamily, and shows a closer phylogenetic relationship to the prokaryotic homodimeric NAD(P)-IDH, while distinguishing itself from them. This implies that UmIDH is not a typical NAD-IDH or NADP-IDH.

A secondary structure-based amino acid sequence alignment was performed to speculate the residues of Type I homodimeric IDHs involved in coenzyme binding (Figure 2), the result indicated that the putative critical coenzyme binding sites of UmIDH differ significantly from its homologue. Among the Type I homodimeric IDHs analyzed, it has been consistently observed that NAD-IDHs utilize Asp, Ile, and Ala residues for binding to NAD^+^ and NADP-IDHs utilize Lys, Tyr, and Val residues for binding to NADP^+^. Dual coenzyme-dependent MfIDH’s corresponding homologous sites are Lys, Ile, and Ala, and binding sites for UmIDH are Arg, Leu, and Val.

### 2.2. Expression and Purification

The recombinant UmIDH with a 6 × His-tag was effectively expressed in *E. coli* Rosetta (DE3) and subsequently purified using Co^2+^ affinity chromatography. SDS-PAGE analysis revealed that the molecular mass of UmIDH was approximately 45 kDa (Figure 3A), which closely matched the theoretical value of 46.7 kDa. Gel filtration chromatography analysis was used to determine the oligomeric status of UmIDH in solution, and a single symmetric elution peak with a molecular mass of 84.7 kDa was observed, indicating that the UmIDH exists as a homodimer in solution (Figure 3B).

### 2.3. The Effects of pH, Temperature, and Metal Ions

The effects of pH and temperature on the activity of UmIDH were evaluated using NADP^+^ as a coenzyme. The optimum pH of UmIDH was approximately 8.5 with Mn^2+^ and 8.7 with Mg^2+^ (Figure 4A). This is similar to the optimum pH of NAD-ZmIDH (pH 8.0 with Mn^2+^ and pH 8.5 with Mg^2+^) [12], lower than that of NAD-*H. thermophilus* IDH (pH 10.5 with Mg^2+^) [20], but higher than that of NADP-MfIDH (pH 6.0 with Mn^2+^) [17]. Recombinant UmIDH exhibited excellent stability in alkaline conditions, maintaining over 60% of its maximum activity across a broad pH range of 7.5 to 9.0, and the alkaline nature of UmIDH may be related to the marine living environment of *U. marinipuiceus*.

The influence of temperature on enzyme activity is manifested in two characteristics: firstly, enzymes have an optimal temperature, and secondly, enzymes have a range of temperature tolerance. UmIDH exhibited maximal activity at approximately 35 °C in the presence of Mn^2+^ or Mg^2+^ (Figure 4B), similar to that of marine bacterial IDHs from *Congregibacter litoralis* (35 °C with Mn^2+^ or Mg^2+^) [21] and *Psychromonas marina* (35 °C with Mn^2+^) [22], but lower than that of ZmIDH (55 °C with Mn^2+^ or Mg^2+^) [12] and SsIDH (50 °C with Mg^2+^) [11]. The heat-inactivation profiles suggested that UmIDH can maintain over 60% of its activity before 50 °C (Figure 4C), while most marine bacterial IDHs have already completely lost their activity at this temperature after a 20 min incubation [21,22,23].

The impact of nine different metal ions on the activity of UmIDH were assessed in the NADP^+^-linked reaction (Table 1). The results indicated that UmIDH, like other previously studied IDHs, relied entirely on the presence of a divalent cation. Among the activators tested for UmIDH catalysis, Mn^2+^ emerged as the most effective, while Mg^2+^ was almost as effective, capable of partially substituting for Mn^2+^ activation (19.51%). In the presence of Mn^2+^ or Mg^2+^, three divalent metal cations (Cu^2+^, Co^2+^, and Ca^2+^) exhibited inhibitory effects on the activity of UmIDH.

### 2.4. Kinetics Analysis

The recombinant UmIDH demonstrated a highly similar affinity for and catalytic efficiency with either coenzyme, indicating UmIDH is a dual coenzyme-dependent IDH, although it slightly prefers NADP^+^. The *K*_m_ values of UmIDH for NAD^+^ and NADP^+^ were 1800.0 ± 64.4 μM and 1167.7 ± 113.0 μM, respectively, while the catalytic efficiency (*k*_cat_/*K*_m_) was only 2.3-times higher for NADP^+^ than for NAD^+^ (Table 2). UmIDH exhibits lower enzymatic activity compared to other type I homodimeric IDHs, for example, NAD-AtIDH (*k*_cat_/*K*_m_ = 0.25 μM^−1^s^−1^) [24], NAD-*S. mutans* IDH (*k*_cat_/*K*_m_ = 0.36 μM^−1^s^−1^) [25], NAD-*P. furiosus* IDH (*k*_cat_/*K*_m_ = 0.88 μM^−1^s^−1^) [26], NADP-EcIDH (*k*_cat_/*K*_m_ = 4.7 μM^−1^s^−1^) [27], and NADP-*M. aeruginosa* IDH (*k*_cat_/*K*_m_ = 0.8 μM^−1^s^−1^) [28].

By aligning UmIDH with typical Type I homodimeric NAD-IDHs and NADP-IDHs, three residues (Arg345, Leu346, and Val352) were identified as putative coenzyme binding sites for UmIDH (Figure 2). In this study, two mutants (a double mutant R^345^K/L^346^Y and a triple mutant R^345^D/L^346^I/V^352^A) were constructed for bidirectional modification of the coenzyme specificity of UmIDH. Circular dichroism spectra (Figure 4D) revealed that the mutant enzymes retained their secondary structure after mutagenesis.

The kinetic parameters of the UmIDH mutants are presented in Table 2. The data suggested that, compared to the wild-type UmIDH, R^345^K/L^346^Y had a *K*_m_ value that was approximately 2-times lower and a catalytic efficiency towards NADP^+^ that was 2.6-times higher. In contrast, R^345^D/L^346^I/V^352^A showed a decrease (approximately 4.9-fold) in the *K*_m_ value and an increase (about 11.4-fold) in the catalytic efficiency towards NAD^+^, when compared to the wild-type UmIDH.

## 3. Discussion

The IDH family, as housekeeping proteins within organisms, has ancient origins, widespread distribution, and species-diversity, making it a valuable basis for studying protein molecular evolution. In this study, the Type I homodimeric UmIDH was characterized. From the phylogenetic tree (Figure 1), it can be observed that although UmIDH clusters with the prokaryotic homodimeric IDH, it is distinct from the NAD-IDH and NADP-IDH branches, indicating its dual coenzyme specificity. The determination of kinetic parameters confirmed that UmIDH is indeed a dual coenzyme-dependent IDH.

The concept of dual coenzyme-dependent IDH has been proposed in the past; however, these IDHs exhibit a significant difference in their preference for one of the two coenzymes [14,15,16]. Therefore, strictly speaking, these IDHs could not be referred to as dual coenzyme-dependent IDHs. Subsequent studies have discovered some IDHs that exhibit the ability to utilize both coenzymes; however, they similarly cannot be called dual coenzyme-dependent IDHs [7,29]. MfIDH is a recently reported dual coenzyme-dependent IDH that shows similar affinity for both NAD^+^ and NADP^+^. Compared to MfIDH, the UmIDH reported in this study better fits the characteristics of dual coenzyme dependency. This is evident from two findings: first, UmIDH exhibits more similar catalytic efficiency between the two coenzymes (the ratio of catalytic efficiency between the two coenzymes is 1:2 for UmIDH, while for MfIDH, this ratio is 1:5), and second, the conservation of the coenzyme binding sites in UmIDH is lower.

Type I IDH, Type II IDH, and Type III IDH have a monophyletic origin and encompass both NAD-IDH and NADP-IDH [7,21]. The results of molecular phylogenetic studies indicate that approximately 3.5 billion years ago, bacterial IDH solely relied on NAD^+^ as its coenzyme. Subsequently, certain bacteria encountered environmental stress due to carbon source limitations, leading to a coenzyme specificity shift in IDH from NAD^+^ to NADP^+^, and this transition provided the bacteria with the necessary reducing power, NADPH. The coenzyme adaptive evolution mechanism of IDH has been reproduced and validated in *E. coli* [10,13,30,31], and numerous experiments have demonstrated its applicability across different subfamilies [21,32,33,34,35]. The majority of the existing NAD-IDHs are found in the Type I subfamily, while a small number are also present in the Type II and Type III subfamilies. For example, marine bacterial homohexameric NAD-IDH and eukaryotic homodimeric NAD-IDH are considered the ancestors of Type II IDHs [21,29,32], while *Campylobacter* NAD-IDHs are considered the ancestors of Type III IDHs [7,29].

Each subfamily of IDH has two sets of amino acid residues for binding different coenzymes, and these residues are highly conserved across IDHs of different subfamilies. In Type I subfamily, different oligomeric forms of IDHs utilize Asp, Ile, and Ala in the coenzyme binding pocket to bind NAD^+^, while Lys, Tyr, and Val are utilized for NADP^+^ binding. In UmIDH, the corresponding homologous site is replaced with Arg, Leu, and Val, which differs from the typical coenzyme binding sites of Type I IDHs and results in the dual coenzyme specificity of UmIDH. We speculate that the coenzyme determinant Asp, Ile, and Ala represents the ancestral phenotype, Lys, Tyr, and Val represents the evolved phenotype, and Arg, Leu, and Val represents an intermediate state between the two. It can be anticipated that the intermediate state phenotype cannot be well stabilized due to its lower catalytic activity; therefore, UmIDH is a survivor.

Early studies have shown that EcIDH is regulated by phosphorylation through the IDH kinase/phosphatase (IDH K/P) system, resulting in a loss of 75–80% of its activity in carbon-limited conditions. This allows the glyoxylate shunt, facilitated by the enzyme isocitrate lyase (ICL), to competitively bind and utilize more substrates [36,37]. In the *E. coli* genome, the gene *aceK*, which encodes EcIDH K/P, is part of the *aceBAK* operon along with the genes *aceA* and *aceB*, which encode ICL and malate synthase (MS), respectively [38,39]. Therefore, the expression of IDH K/P is coupled with the expression of ICL and MS, both of which are key enzymes in the glyoxylate bypass pathway. Through bioinformatic analysis, we discovered that the *M. flagellatus* genome lacks the *aceBAK* operon, while the *U. marinipuiceus* genome possesses a complete *aceBAK* operon, which is consistent with *E. coli*, this indicates that UmIDH has a physiological basis for phosphorylation, and the potential phosphorylation site is Ser114 (Appendix A). We also analyzed the potential acetylation and succinylation sites of UmIDH and found that it has fewer relevant sites compared to EcIDH and MfIDH (Appendix A) [17,40,41].

Bioinformatics analysis methods and protein engineering approaches play a crucial role in the field of molecular evolution. The former involves comparing enzyme sequences with sequence information in databases, enabling the prediction of enzyme structure and function. Through sequence conservation analysis, specific sequence patterns related to substrate specificity can be identified. The latter allows for the alteration of substrate specificity through methods such as site-directed mutagenesis, insertion, deletion, or recombination. Additionally, protein engineering methods utilize directed evolution, enzyme library screening, and computational simulations to design and optimize enzyme catalytic performance and substrate specificity [42,43]. One method to validate an evolutionary pathway is reverse engineering. In this study, two mutants were constructed, and only two to three sites mutations are required to achieve bidirectional modification of the coenzyme specificity of UmIDH, which indicates that our speculation is reasonable. This current work provide compelling evidence for the evolutionary mechanism of coenzyme specificity of IDH.

## 4. Materials and Methods

### 4.1. Strains, Plasmids and Reagents

*E. coli* DH5α and *E. coli* Rosetta (DE3) were stored at an ultra-low temperature freezer for preservation. Full-length gene encoding UmIDH was synthesized by Generay Biotech Co., Ltd. (Shanghai, China), and the coding sequence was codon optimized according to the *Escherichia coli* bias. The plasmid used was pET28b, and the restriction enzyme sites at both ends of the gene were *Nde*I and *Xho*I. PrimeSTAR Max DNA Polymerase was purchased from TaKaRa (Dalin, China). Restriction enzymes (*Nde*I, *Xho*I, and *Dpn*I), T4 DNA ligase, and Protein Molecular Weight Standards were purchased from Thermo Fisher Scientific (Shanghai, China). The TALON Metal Affinity Resin used for protein purification was obtained from TaKaRa. All other biochemical reagents were sourced from Sangon Biotech and BBI (Shanghai, China).

### 4.2. Site-Directed Mutagenesis

UmIDH mutants (R^345^K/L^346^Y and R^345^D/L^346^I/V^352^A) were constructed through site-directed mutagenesis by using UmIDH-pET28b as the template. The primers that were used to generate the R^345^K/L^346^Y mutant were as follows: forward, 5′-CATGGTACTGCCCCTAAATATGCAGGTATGGATC-3′, and reverse, 5′-ATATTTAGGGGCAGTACCATGGGTCGGTTCA-3′. The R^345^D/L^346^I/V^352^A was constructed by two rounds of site-directed mutagenesis. The first round of mutation introduced the R^345^D/L^346^I change, using the following primers: forward, 5′-CATGGTACTGCCCCTGATATTGCAGGTATGGAT-3′, and reverse, 5′-AATATCAGGGGCAGTACCATGGGTCGGTTC-3′. The second round of mutation introduced the V^352^A change, using the following primers: forward, 5′-GCAGGTATGGATCGTGCAAATCCGTGTAGC-3′, and reverse, 5′-TGCACGATCCATACCTGCAATATCAGGGG-3′. The underlined codons are mutated sequences. After PCR, the template sequences were eliminated using the methylation-specific *Dpn*I enzyme. Then, the products were transformed into competent bacterial strains.

### 4.3. Recombinant Protein Expression and Purification

*Escherichia coli* Rosetta (DE3) strains harboring the recombinant plasmid were pre-cultured overnight in Luria-Bertani (LB) medium supplemented with 30 μg/mL kanamycin and 30 μg/mL chloramphenicol at 37 °C with agitation at 225 rpm. Subsequently, the cells were inoculated at a 1:100 dilution into 200 mL fresh LB medium containing the same antibiotics and incubated at 37 °C and 225 rpm. The expression of the recombinant protein was induced by adding 0.5 mM IPTG when the optical density of the cultures reached an OD_600_ of 0.4–0.6. The cultivation was continued for 20 h at 20 °C with agitation at 180 rpm. The cells were harvested by centrifugation at 5000 rpm for 5 min at 4 °C. The resulting pellets were then resuspended in lysis buffer containing 20 mM Tris-HCl, 300 mM NaCl (pH 7.5). After sonication, the cell debris was eliminated by centrifugation at 10,000 rpm for 20 min at 4 °C. The supernatant containing the 6×His-tagged recombinant UmIDH was purified Co^2+^ affinity resin (Cat. #635502, Clontech, TaKaRa, Dalin, China).

### 4.4. SDS-PAGE and Gel Filtration Chromatography

In order to determine the molecular mass of the subunit(s), the purified proteins were subjected to denaturing electrophoresis using a discontinuous SDS-PAGE system with a 5% stacking gel and a 12% separating gel. The molecular mass of the active UmIDH was estimated using gel filtration chromatography on the Superdex 200 10/300 increase column (GE Healthcare Life Sciences, Pittsburgh, PA, USA), which was equilibrated with a 50 mM potassium phosphate buffer (pH 7.0) containing 150 mM NaCl and 0.01% sodium azide. Protein standards such as Ovalbumin (44 kDa), Conalbumin (75 kDa), Aldolase (158 kDa), Ferritin (440 kDa), and Thyroglobulin (669 kDa) were used for calibration of the gels.

### 4.5. Circular Dichroism Spectroscopy

Circular dichroism (CD) spectra were acquired using a Jasco model J-810 spectropolarimeter. UmIDH and mutants concentrations were diluted to 0.2 mg/mL using a CD buffer containing 20 mM NaH_2_PO_4_ and 75 mM Na_2_SO_4_ at pH 7.5. Then, 200 μL of the protein was loaded into a microcuvette and transferred into the instrument at a wavelength range of 190 nm to 280 nm. The mean residue ellipticity ([*θ*], deg·cm^2^·dmole^−1^) of the protein was calculated using the following formula [*θ*] = *θ*/10·(*n* − 1)·*C*·l, where *θ* is the ellipticity measured in mdeg; n is the number of protein amino acid residues; *C* is the concentration of the protein in mol/L; and l is the light path of the cuvette (0.1 cm). Three scans were performed for each sample, and the average value was recorded. The protein secondary structure was estimated following the method described by Raussens et al. [44].

### 4.6. Enzyme Characterization

The pH-dependent activity of the recombinant UmIDH was investigated in 50 mM Tris-HCl buffer ranging from pH 7.7 to 9.5 at 25 °C. The temperature dependence of the recombinant UmIDH activity was assessed across the temperature range of 25–60 °C at pH 8.5. The thermostability of the recombinant UmIDH was evaluated by subjecting enzyme aliquots to heat inactivation at temperatures ranging from 25 °C to 65 °C at pH 8.5 for 20 min. Following incubation, the aliquots were rapidly cooled on ice, and the residual enzyme activity was determined using the standard enzyme assay. The effects of various metal ions (Mn^2+^, Mg^2+^, Ca^2+^, Co^2+^, Cu^2+^, Ni^2+^, K^+^, Na^+^, and Li^+^) on MmIDH were assessed using the standard assay method. All data for enzyme activity were tested in at least three independent experiments.

### 4.7. Enzyme Assay

Enzyme assays were performed in a 1 mL reaction volume consisting of 50 mM Tris-HCl (pH 8.5), 2 mM MnCl_2_ or MgCl_2_, 1 mM trisodium DL-isocitrate, and 0.5 mM NAD(P)^+^ at a temperature of 25 °C. The increase in NAD(P)H absorbance was monitored at 340 nm using a Cary 300 UV-Vis spectrophotometer (Varian, Polo Alto, CA, USA) with an extinction coefficient (ε340) of 6.22 mM^−1^·cm^−1^. One unit (U) of enzyme activity was defined as the generation of 1 μmol of NADPH per minute. Protein concentrations were determined using a Bio-Rad protein assay kit (Bio-Rad, Hercules, CA, USA). The enzyme’s kinetic parameters were determined by measuring its activity at various concentrations of NADP^+^ or NAD^+^. The apparent kinetic parameters were calculated using nonlinear regression analysis with Prism 5.0 software (GraphPad Software, San Diego, CA, USA). All experiments were conducted at least three times to ensure reproducibility.

## Figures and Tables

**Figure 1 ijms-24-11428-f001:**
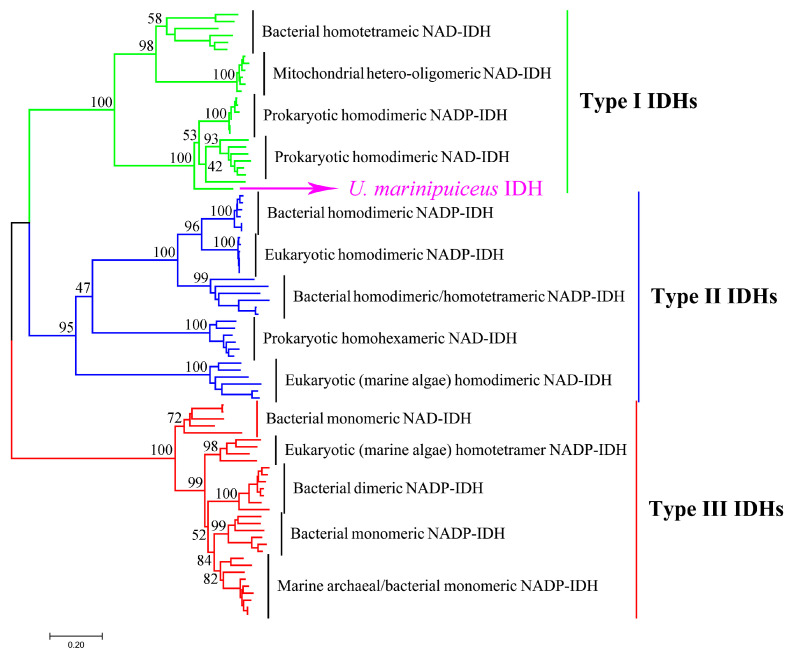
The phylogenetic tree of IDH family. The evolutionary tree was constructed by MEGA 7.0 using the neighbor-joining method with 1000 bootstrap replicates, which involved 87 IDH sequences. The UmIDH was targeted by pink arrow.

**Figure 2 ijms-24-11428-f002:**
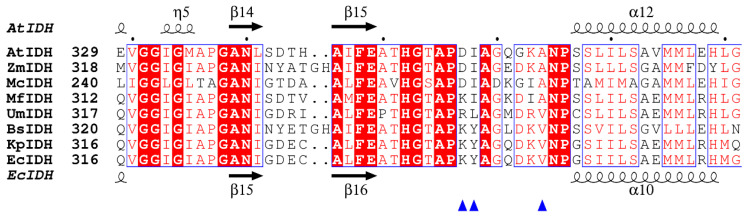
Structure-based amino acid sequence alignment. The comparison of UmIDH with three typical Type I homodimeric NAD-IDHs from *Acidithiobacillus thiooxidans* (AtIDH, NCBI Reference Sequence: WP_193650900.1), *Zymomonas mobilis* (ZmIDH, NCBI Reference Sequence: WP_014500849.1) and *Methylococcus capsulatus* (McIDH, NCBI Reference Sequence: WP_010962255.1), and three typical Type I homodimeric NADP-IDHs from *Escherichia coli* (EcIDH, NCBI Reference Sequence: WP_096963047.1), *Bacillus subtilis* (BsIDH, NCBI Reference Sequence: WP_003152454.1), and *Klebsiella pneumoniae* (KpIDH, NCBI Reference Sequence: WP_129543013.1), and dual coenzyme-dependent MfIDH (NCBI Reference Sequence: WP_011480360.1). The structure of AtIDH (PDB ID: 2D4V) and EcIDH (PDB ID: 3ICD) were downloaded from the PDB database. The critical coenzyme binding sites were indicated by blue triangles. The figure was created by ESPript 3.0.

**Figure 3 ijms-24-11428-f003:**
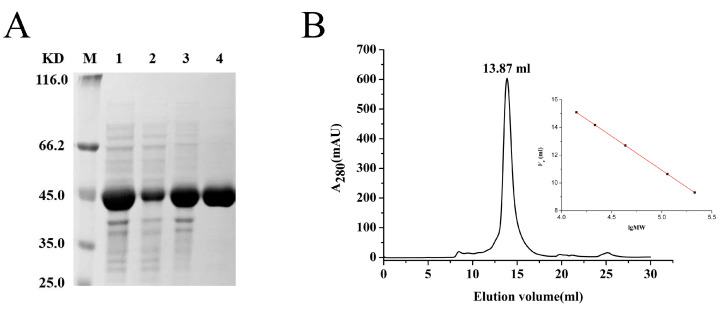
Overexpression and purification of UmIDH. (**A**) SDS-PAGE analysis. M, molecular mass marker; lane 1, crude extract; lane 2, supernatant of crude extract; lane 3, precipitation of crude extract; lane 4, purified UmIDH. (**B**) Gel filtration chromatography analysis. *V*_e_ of the UmIDH was 13.87 mL.

**Figure 4 ijms-24-11428-f004:**
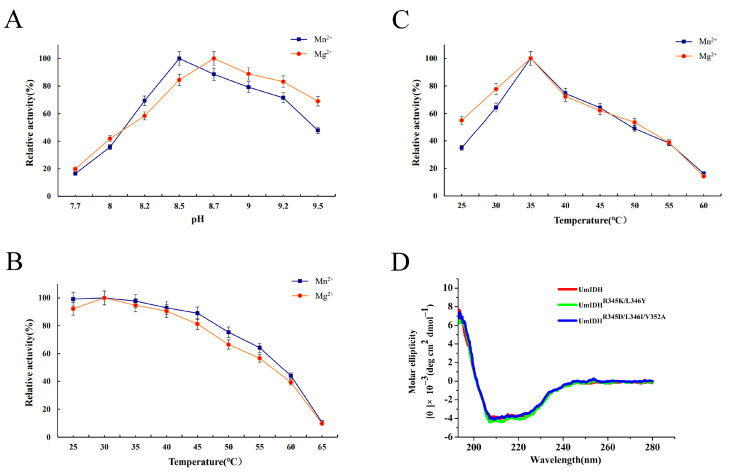
(**A**) Effects of pH on the activity of UmIDH in the presence of Mn^2+^ or Mg^2+^. (**B**) Effects of temperature on the activity of UmIDH in the presence of Mn^2+^ or Mg^2+^. (**C**) Heat-inactivation profiles of UmIDH in the presence of Mn^2+^ or Mg^2+^. (**D**) CD spectra of the wild-type UmIDH and its mutants.

**Table 1 ijms-24-11428-t001:** Effect of different metal ions on the activity of UmIDH.

Metal Ions	Relative Activity (%)	Metal Ions	Relative Activity (%)	Metal Ions	Relative Activity (%)
None	0				
Mn^2+^	100 ± 0	Mn^2+^	100 ± 0		
Mg^2+^	19.51 ± 4.66	Mn^2+^ + Mg^2+^	114.97 ± 5.13	Mg^2+^	100 ± 0
Ca^2+^	15.90 ± 1.32	Mn^2+^ + Ca^2+^	67.10 ± 4.27	Mg^2+^ + Ca^2+^	20.75 ± 1.34
Co^2+^	0.66 ± 1.29	Mn^2+^ + Co^2+^	20.04 ± 2.80	Mg^2+^ + Co^2+^	7.68 ± 0.74
Cu^2+^	0.31 ± 0.27	Mn^2+^ + Cu^2+^	4.04 ± 0.33	Mg^2+^ + Cu^2+^	6.97 ± 0.61
Ni^2+^	2.29 ± 0.64	Mn^2+^ + Ni^2+^	103.51 ± 3.42	Mg^2+^ + Ni^2+^	102.07 ± 0.59
K^+^	8.30 ± 0.43	Mn^2+^ + K^+^	106.13 ± 4.37	Mg^2+^ + K^+^	110.91 ± 1.10
Na^+^	2.31 ± 0.75	Mn^2+^ + Na^+^	104.88 ± 3.92	Mg^2+^ + Na^+^	101.03 ± 2.11
Li^+^	7.69 ± 1.07	Mn^2+^ + Li^+^	103.53 ± 0.98	Mg^2+^ + Li^+^	107.04 ± 1.20

The relative activity was assessed in a standard reaction mixture containing 2 mM of the indicated metal ion(s). Data are the mean ± SD of at least three independent measurements.

**Table 2 ijms-24-11428-t002:** The kinetic parameters of the wild-type UmIDH and its mutants.

Enzyme		NAD^+^			NADP^+^	
	*K*_m_ (μM)	*k*_cat_ (s^−1^)	*k*_cat_/*K*_m_ (μM^−1^s^−1^)	*K*_m_ (μM)	*k*_cat_ (s^−1^)	*k*_cat_/*K*_m_ (μM^−1^s^−1^)
UmIDH	1800.0 ± 64.4	7.5 ± 0.3	0.0042	1167.7 ± 113.0	11.1 ± 0.4	0.0095
R345K/L346Y	-	-	-	605.9 ± 38.2	15.3 ± 0.2	0.025
R345D/L346I/V352A	369.7 ± 11.84	17.7 ± 0.9	0.048	-	-	-

“-” not detected. The values indicate the means of at least three independent measurements.

## Data Availability

The original contributions presented in the study are included in the article and its Supplementary Material. Further inquiries can be directed to the corresponding authors.

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
