# Peer review of "Enzymatic Characterization of the Isocitrate Dehydrogenase with Dual Coenzyme Specificity from the Marine Bacterium Umbonibacter marinipuiceus"

_ijms, 2023, doi:10.3390/ijms241411428_

Round 1

Reviewer 1 Report

The paper includes some characterization of isocitrate dehydrogenase from Umbonibacter marinipuiceus, and compares the kinetic values with other bacteria enzymes.

1) In Figure 3B, the axis of inserted Figure are exchanged. Volume should be the X-axis, as the value 13,87 mL is not included at the Y-axis, where Volume is written. Please, change the labels in X, Y.

2) Figure 4A.- I have not found the temperature used in the pH plot. I also didn't find in the Methods section the temperature used. Is it 35 C?

3) Figure 4B and Figure 4C.- The same regarding pH used in both plots. Which is the pH used?

4) Table 1.- I cannot see which were the concentrations used for all the ions. Specially interesting is the [Mn2+] and [Mg2+] when two different ions were used (rows 3 and 4 and rows 5 and 6). Was the ionic strengh manteined in all the buffers?

5) Line 81.- It should be "analysis" instead of "nanlysis"

For maquetation: Line 242 is not good maqueted / Line 256 should change to the following page.

I think English Language is undestandable. I undertood what authors explained in the paper.

Reviewer 2 Report

I have reviewed the manuscript titled "Enzymatic characterization of the isocitrate dehydrogenase with dual coenzyme specificity from the marine bacterium Umbonibacter marinipuiceus." The authors aim to elucidate the properties of NAD- and NADP-dependent dual isocitrate dehydrogenase (IDH) and have created NAD-specific and NADP-specific IDH mutants based on the enzyme's amino acid sequence.

The experimental work carried out in this study is well-conducted and comprehensive. The authors collected a considerable amount of data, which contributes to our understanding of the fundamental properties of the enzyme. However, I have concerns regarding the novelty of this research.

The manuscript mainly consolidates previously known experimental results and lacks significant new findings. As a result, it falls short in terms of making a substantial impact on the field. To warrant publication in IJMS, there must be a stronger emphasis on the novelty aspect of this research. I suggest the authors reconsider their approach and highlight the unique aspects of their study, clearly stating how it advances the current understanding of NAD- and NADP-dependent dual IDHs.

Without a clear demonstration of novelty and the advancement of knowledge in the field, I cannot recommend accepting this paper for publication in IJMS at this point.

I hope my feedback will be helpful to the authors in revising their manuscript.

Reviewer 3 Report

ijms-2504730

Title - Enzymatic characterization of the isocitrate dehydrogenase with dual coenzyme specificity from the marine bacterium Umbonibacter marinipuiceus.

The manuscript by Bian et al. demonstrated dual coenzyme-dependent IDH expressed, purified, and characterization in brief. Overall, the manuscript requires major revision before its publication in “IJMS” as follows:

 Comments:

1.      The authors should follow the significant number rule to present data in abstract, text, and Tables.

2.      Lines 41-50, please add some additional information on coenzyme specificity/selectivity for an enzyme with examples and the role/significance of protein engineering approaches, including bioinformatics for validation catalytic activity/selectivity/dual substrate specificity i.e., https://doi.org/10.1016/j.ijbiomac.2018.11.096; https://doi.org/10.3390/ijms21217859. Also, discussed such strategies’ importance in the discussion section.

3.      Figure 1. The organism’s name should be in italics, and font size should be improved.

4.      Figure 4 (A-C) data can be presented in terms of specific activity (U/mg of protein) instead of relative activity.

5.      Table 1 results i.e., a significant alteration of activity in the presence of bi- metals should be instrumentally validated and justified with evidence.

6.      Discussions are weak, it requires significant improvements to justify the significance of this study’s experimental findings.  

Minor editing of English language is required.

Round 2

Reviewer 2 Report

I recommend accepting this manuscript for publication for IJSM as the previously highlighted issues have been adequately addressed and improved.

Reviewer 3 Report

Accept as is